# Optimising the DPPH Assay for Cell-Free Marine Microorganism Supernatants

**DOI:** 10.3390/md19050256

**Published:** 2021-04-29

**Authors:** Yehui Gang, Tae-Yang Eom, Svini Dileepa Marasinghe, Youngdeuk Lee, Eunyoung Jo, Chulhong Oh

**Affiliations:** 1Jeju Marine Research Center, Korea Institute of Ocean Science and Technology, 2670 Iljudong-ro, Gujwa-eup, Jeju-si 63349, Korea; yehui@kiost.ac.kr (Y.G.); eomsun@kiost.ac.kr (T.-Y.E.); svini91@kiost.ac.kr (S.D.M.); lyd1981@kiost.ac.kr (Y.L.); jey8574@kiost.ac.kr (E.J.); 2Department of Ocean Science, University of Science and Technology, 217 Gajeong-ro, Yuseong-gu, Daejeon 34113, Korea

**Keywords:** DPPH assay, antioxidant activity, marine microorganism, cell-free supernatant, salt

## Abstract

Antioxidants prevent ageing and are usually quantified and screened using the 1,1-diphenyl-2-picrylhydrazyl (DPPH) assay. However, this assay cannot be used for salt-containing samples, such as the cell-free supernatants of marine microorganisms that are aggregated under these conditions. Herein, the DPPH solvent (methanol or ethanol) and its water content were optimized to enable the analysis of salt-containing samples, aggregation was observed for alcohol contents of >70%. The water content of methanol influenced the activities of standard antioxidants but did not significantly affect that of the samples. Based on solution stability considerations, 70% aqueous methanol was chosen as the optimal DPPH solvent. The developed method was successfully applied to the cell-free supernatants of marine bacteria (*Pseudoalteromonas rubra* and *Pseudoalteromonas xiamenensis*), revealing their high antioxidant activities. Furthermore, it was concluded that this method would be useful for the screening of marine microorganism–derived antioxidants, which also has numerous potential applications, such as salt-fermented foods.

## 1. Introduction

Antioxidants inhibit the decay of certain substances, reduce oxidative stress, and hinder cellular ageing [1]. Therefore, they have numerous applications in pharmaceutical, cosmetic, and food industries [2]. Despite the abundance of synthetic antioxidants, concerns related to their toxicity and side effects have inspired the search for natural antioxidants [3]. Recently, extensive research has been conducted on the screening of natural antioxidants from a variety of sources, such as plants, animals, moss, mushrooms, fungi, bacteria, and turquoise algae [4]. It is industrially advantageous to produce useful substances using microorganisms, as this approach is inexpensive and enables ease of extraction, strain improvement, and environmental friendliness [5,6]. Consequently, antioxidants produced by microorganisms have recently drawn much attention [7,8] and have been isolated from cell-free supernatants [9] as many microorganisms secrete metabolites to the surrounding environment [10]. Pienze et al. showed that the antioxidant activities of cell-free supernatants of probiotic bacteria *Enterococcus durans* LAB18s exceeded those of intracellular extracts when analyzed by ABTS and DPPH assays [11].

Marine microorganisms produce a larger variety of bioactive compounds than terrestrial ones, as the former grow in harsh environments characterized by high salt contents, high pressures, and low temperatures [12,13]. Moreover, the high genetic diversification in microorganisms isolated from the marine environment has contributed to the production of novel metabolites with beneficial functions [14]. However, the antioxidants contained in the supernatants of marine microorganisms have been studied much less than those contained in the supernatants of terrestrial microorganisms. Therefore, research using the cell-free supernatant of marine microbes is needed to produce new or more effective antioxidants.

In order to evaluate the antioxidant capacity of cell-free supernatants, it is necessary to consider an appropriate analytical method depending on the characteristics of the supernatant [15]. Among the various analytical methods, the scavenging of the stable 1,1-diphenyl-2-picrylhydrazyl (DPPH) radical is a fast, simple, and economical method for determining the antioxidant activity [16]. In previous studies, this process has been successfully used to measure the antioxidant activity of microorganisms and has been widely applied not only to cultures, but also to extracts (Table 1). The DPPH analysis can be applied to confirm the antioxidant activity of supernatants of marine microorganisms as they are in a state in which specific components are not separated and various substances such as saccharide and pigment are comprehensively dissolved in an aqueous solution [17]. Although this absorbance measurement–based assay can be modified to analyze various substances at diverse concentrations and under various conditions, it is not suited for substances (e.g., proteins) that can precipitate in alcohol-based solvents [18]. Similarly, for solutions containing salts, the results can be affected by precipitation caused by the alcoholic component of the DPPH solution [19,20]. For this reason, it is difficult to measure the antioxidant activity of samples containing salts, such as the cell-free supernatant of marine microorganisms. 

Therefore, in this study, the concentration of alcoholic solvents that were not affected by salt was identified and used to measure the DPPH antioxidant activity. In this sense, we confirmed the applicability to the cell-free supernatant of marine microorganisms.

## 2. Results

The aggregation behavior of marine broth (MB) and artificial seawater was determined after the addition of DPPH solvent, methanol (MeOH), and ethanol (EtOH), with different alcoholic contents. The salinity of MB was 32 psu and the same salinity was used for the preparation of artificial seawater. Aggregation was observed for all alcohol contents except for 60% and 70% (Table 2 and Figure 1). The immediate reaction was observed after the addition of alcoholic solvents and the precipitate was settled at the bottom of the well. The precipitate formed by the reaction between 80% MeOH with MB affects the absorbance although the aggregation was weaker than the other experimental groups. The concentration below 60% was not considered as the DPPH did not completely dissolve at MeOH and EtOH levels of ≤50% (data not shown). Thus, a DPPH solution with 70% MeOH was used for the upcoming experiments in this study to determine the antioxidant activities of standard antioxidants and salt containing samples.

The influence of 70% MeOH in the DPPH solution on the antioxidant activities of antioxidants were investigated (Figure 2). l-ascorbic acid and butylated hydroxytoluene (BHT) which are representative antioxidants widely used in the food industry were used as standard antioxidants [4]. The results obtained from the experiment suggest that there is no significant difference between the scavenging activities of l-ascorbic acid when using 70% and 100% MeOH in DPPH solutions (Figure 2a). Similarly, a significant difference was not observed for the antioxidant activities of over the entire concentration range of BHT when using 70% and 100% MeOH in DPPH solutions (Figure 2b). Thus, the results implied that an application of DPPH assay is possible with the DPPH solution containing 70% MeOH.

The effects of salt on the antioxidant activity were determined by probing solutions of l-ascorbic acid in distilled water and artificial seawater (32 psu) using a DPPH solution in 70% MeOH (Figure 3). The experiment was not performed with BHT as it was a hydrophobic antioxidant. At l-ascorbic acid concentrations of 0.2 mg/mL, no difference was observed, whereas the antioxidant activities of the distilled water solution exceeded that of the seawater solution about 10% and 5% at a concentration of 0.1 and 0.15 mg/mL, respectively. However, when comparing the two groups, there was no statistically significant difference at the 95% significance level. Therefore, it suggests the possible use of the DPPH solution containing 70% MeOH to measure the antioxidant activities of salt containing solutions.

Based on the results obtained from the experiments mentioned above, the solution of DPPH in 70% MeOH was used to determine the antioxidant activities of the cell-free supernatants of *Pseudoalteromonas rubra* and *Pseudoalteromonas xiamenensis.* DPPH radical scavenging activities of supernatants of *P. rubra* in MB and SM (seawater medium) were higher than that of *P. xiamenensis.* Cell-free supernatants of *P. rubra* obtained at a 24 h incubation time showed the highest scavenging activities of 52% and 76% in MB and SM, respectively (Figure 4a). The scavenging activity of 44% was observed for the supernatant of *P. xiamenensis* in SM collected at 48 h, while the supernatants of *P. xiamenensis* cultured in MB showed the low scavenging activities (Figure 4b). According to the data, the supernatants obtained from SM showed the higher overall scavenging activities than that of supernatants of MB.

## 3. Discussion

The DPPH assay is useful for investigating the antioxidant properties of extracts obtained using various solvents, is cheap and easy to implement, and can be applied to both hydrophilic and lipophilic antioxidants [18]. In addition, this assay has been successfully used to measure the antioxidant activities of carotenoids, polysaccharides, and MeOH extracts of marine microorganisms [24,25,26]. Therefore, this method can be chosen as an effective method for comprehensively screening the antioxidant effects of supernatant containing various substances. However, in its current version, the DPPH assay is not suitable for salt-containing samples, as agglutination occurs under these conditions, resulting in precipitation and thus affecting the outcomes of absorbance measurements. Samples containing large amounts of proteins or salts may not be suitable for analysis, as they may form precipitates in alcoholic solvents [18]. To address this problem, here the suitability of several alcoholic solvents were examined regarding the analysis of salt-containing samples, revealing that aggregation was observed when salt-containing samples were brought into contact with aqueous MeOH and EtOH solutions having alcohol contents of ≥80% (Table 2, Figure 1). Notably, DPPH did not completely dissolve in aqueous MeOH and EtOH at alcohol contents of ≤50% (data not shown). Furthermore, although solutions of DPPH in 60% MeOH and 60–70% EtOH exhibited no initial precipitation, their stability decreased after several hours, making them difficult to use (data not shown). Therefore, considering the effects of aggregation and DPPH solution stability, the solution of DPPH in 70% MeOH was concluded to be best suited for determining the antioxidant activities of salt-containing samples. Subsequently, standard antioxidants were used to determine the effects of MeOH content on the antioxidant activity, these effects were found to be minor (Figure 2). Notably, the condition of the DPPH solution was found to be important for the outcome of the assay, as this assay essentially measures the color change due to the DPPH removal by a given antioxidant. Thus, as DPPH is affected by light and temperature, it is most desirable to preserve the absorption of the fresh DPPH solution [27]. As the DPPH assay has been applied to various analytes, it would be possible to modify previously reported methods by changing the employed concentration or solvent [28,29]. Sharma and Bhat (2009) showed that the scavenging inhibition pattern of BHT depends on the DPPH solvent. Therefore, suitable analyte characteristics should be identified to ensure a successful analysis, as shown in the present study. The DPPH solution’s composition should also be appropriately controlled and justified in advance. The time required for the reaction to fully proceed is analyte-dependent [29]. For example, BHT reacts slower than l-ascorbic acid, as can be seen by the naked eye. That is, the addition of l-ascorbic acid immediately decreases the intensity of the purple color of DPPH, whereas the addition of BHT has a slower effect. Figure 2 shows that the maximum activities of both antioxidants exceeded 90%, showing that the 30-min reaction time used in this experiment could therefore be considered appropriate. At the same DPPH concentration, the ethanolic DPPH solution had the same absorbance as the methanolic DPPH solution, offering the benefits of stable solution quality and DPPH solubility. However, both l-ascorbic acid and BHT reacted for more than 30 min in the former case, but the reaction did not occur sufficiently and the antioxidant results were not constant. Therefore, under conditions similar to this experiment, methanol is more suitable as a solvent than ethanol. Previous studies have mentioned the conditions of the DPPH solution or the difference between the pH or temperature of the solution and the type of solvent [30]. However, no studies have compared the optimal conditions of DPPH solution to the antioxidant activity of saline-containing samples to analyze the cell-free supernatant of marine microorganisms. To determine whether salt had an effect on the antioxidant activity, DPPH in 70% MeOH was used to probe solutions of l-ascorbic acid in distilled water and seawater (Figure 3). At the highest l-ascorbic acid concentration (0.2 mg/mL), there was little difference between the two experimental groups. However, differences were observed at low l-ascorbic acid concentrations (0.1 and 0.15 mg/mL). This may have been due to the effects of minerals dissolved in seawater on the rates of the antioxidant reactions.

To test the practical applicability of the proposed method (DPPH in 70% aqueous MeOH), it was applied to salt-containing cell-free supernatants of marine microorganisms. Specifically, two species of *Pseudoalteromonas*, which were expected to produce antioxidants, were selected and cultured in MB and SM, respectively. The antioxidant activities of the cell-free supernatants were found to be dependent on the microorganism, incubation time, and medium. Overall, microorganisms cultured in SM afforded extracts with higher antioxidant activities than those cultured in MB (Figure 4). This suggests that the secretion of antioxidants by microorganisms may depend on the medium. At all incubation times, *P. rubra* supernatants showed higher antioxidant activities than those of *P. xiamenensis*. The highest activity (76%) was observed at a culturing time of 24 h and with SM as the culturing medium (Figure 4). The fact that a certain antioxidant activity was observed even at a culturing time of 0 h was ascribed to the influences of certain components of the culture medium.

In summary, here the optimal solvent and concentration for the DPPH assay of salt-containing samples were identified and successfully applied to cell-free supernatants of marine microorganisms. This revealed that the developed method holds great promise for the screening of antioxidants in salt-containing samples. 

## 4. Materials and Methods

### 4.1. Solution Preparation and Bacterial Strains

MB (Difco^TM^ Marine Broth 2216, BD, Franklin Lakes, NJ, USA) was prepared according to the manufacturer’s manual, and artificial seawater (Sigma-Aldrich, St. Louis, MO, USA) was prepared with a salinity equal to that of MB. In both solutions, salinity was measured using a refractive salinity meter (Master-s/mill, ATAGO, Tokyo, Japan). DPPH, l-ascorbic acid, and BHT were purchased from Sigma-Aldrich (St. Louis, MO, USA). DPPH solutions were prepared in MeOH (Duksan, Kyunggi, Korea, HPLC grade) and EtOH (Duksan, Kyunggi, Korea, HPLC grade). *P. rubra* and *P. xiamenensis* were sourced from the Korea Institute of Ocean Science & Technology (Busan, Korea).

### 4.2. Aggregation Properties

MB and artificial seawater (32 psu), each with a volume of 200 μL, were dispensed in 24-well plates. Then, volumes of 1800 μL of MeOH and EtOH with 60%, 70%, 80%, 90%, and 100% alcoholic contents were added to the wells containing MB and artificial seawater separately.

### 4.3. Effects of Alcohol Content on the Antioxidant Activity

We probed the effect of MeOH content in the DPPH solution on the antioxidant activity. Specifically, 0.2 mM DPPH solutions were prepared in aqueous MeOH with alcohol contents of 70% and 100%. l-ascorbic acid (dissolved in distilled water, 0–0.2 mg/mL, interval = 0.04 mg/mL) and BHT (dissolved in 100% methanol, 0–1 mg/mL, interval = 0.2 mg/mL) were used as standard antioxidants. The DPPH assay was performed according to Prieto (2012) by mixing 180 µL of the DPPH solution and 20 µL of the antioxidant solution [31]. The mixture was reacted for 30 min in the dark at room temperature, and absorbance was measured at 517 nm using a spectrophotometer (Multiskan Go. Thermo Scientific, Finland). All the experiments were performed in triplicate, and the scavenging activity (%) was calculated as 100% × (*A*_i_ − *A*_j_)/*A*_c_, where *A*_i_ is the absorbance of the sample + DPPH, *A*_j_ is the absorbance of the sample + DPPH solvent, and *A*_c_ is the absorbance of the DPPH solvent + DPPH.

### 4.4. Effects of Salt on the Antioxidant Activity

l-ascorbic acid was dissolved in distilled water and artificial seawater to concentrations of 0, 0.05, 0.1, 0.15, and 0.2 mg/mL. DPPH was dissolved in 70% aqueous MeOH to a concentration of 0.2 mM, and the assay was performed as described above.

### 4.5. Antioxidant Activity of the Cell-Free Supernatant of Marine Bacteria

The SM was prepared from natural seawater (Jeju, Korea), filtered through a 0.2-µm membrane filter and supplemented with 0.1% yeast extract (BD, Franklin Lakes, NJ, USA) and 0.5% peptone (BD, Franklin Lakes, NJ, USA) to afford a composition similar to that of MB. Figure 5 shows the steps of the DPPH assay used to determine the antioxidant activity of the cell-free supernatant of marine bacteira. Two marine microorganisms, *P. rubra and P. xiamenensis* were cultured in 4 mL MB and SM for 72 h at 30 °C. A culture volume of 500 μL was collected from each sample after incubation for 12, 24, 48, and 72 h. The culturing media were centrifuged (8000 rpm, 3 min), and the supernatants were separated and subjected to the DPPH assay, which was performed with 70% aqueous methanol as the DPPH solvent. 

### 4.6. Statistical Analysis

All the experiments were performed in triplicate, and data were analyzed using GraphPad Prism version 8 (GraphPad Software, San Diego, CA, USA). The results were expressed as means ± standard deviations. Significant differences between the two groups were compared using the *t*-test and the antioxidant activities of *P. rubra* and *P. xiamenensis* were subjected to one-way analysis of variance, with *p* < 0.05 representing significant differences.

## 5. Conclusions

The DPPH assay has been modified to allow the analysis of salt-containing samples. The optimal DPPH solvent was identified as 70% aqueous methanol. When 70% aqueous methanol was used, l-ascorbic acid and BHT were investigated as standard antioxidants, and there was no difference in the antioxidant activity verification power. In addition, the antioxidant activity was continuously increased regardless of the salt concentration of the l-ascorbic acid aqueous solution. We used a modified method to investigate the cell-free supernatant of marine microorganisms. Through the DPPH analysis, it was confirmed that the supernatant had a high antioxidant activity depending on the culture time and medium. There is a limit to knowing which antioxidants are present in the supernatant using only the DPPH analysis. However, by analysing the supernatant before extracting specific substances, the modified DPPH assay will help discover and identify marine microorganisms that could be developed as antioxidant production tools.

## Figures and Tables

**Figure 1 marinedrugs-19-00256-f001:**
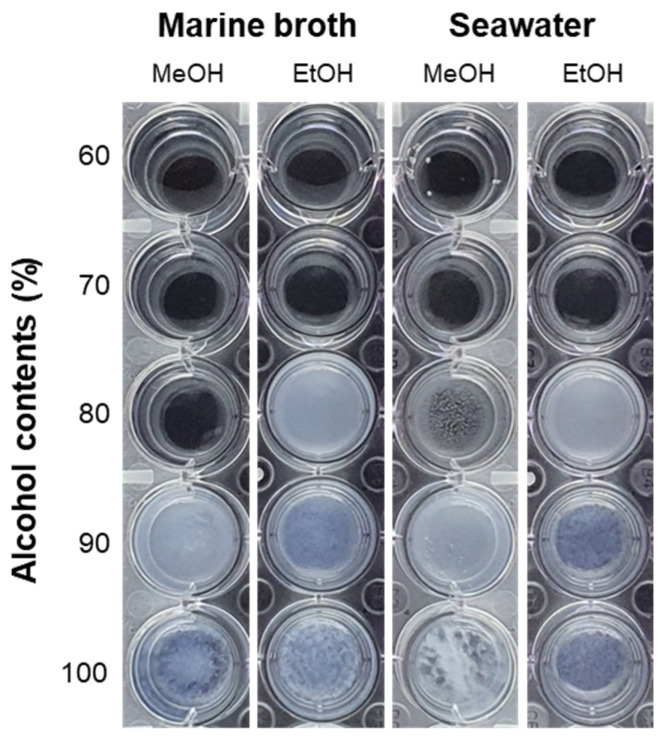
Aggregation behavior observed upon the addition of aqueous MeOH and EtOH with alcohol contents of 60%, 70%, 80%, 90%, and 100% to MB (32 psu) and seawater (32 psu) (v:v = 9:1).

**Figure 2 marinedrugs-19-00256-f002:**
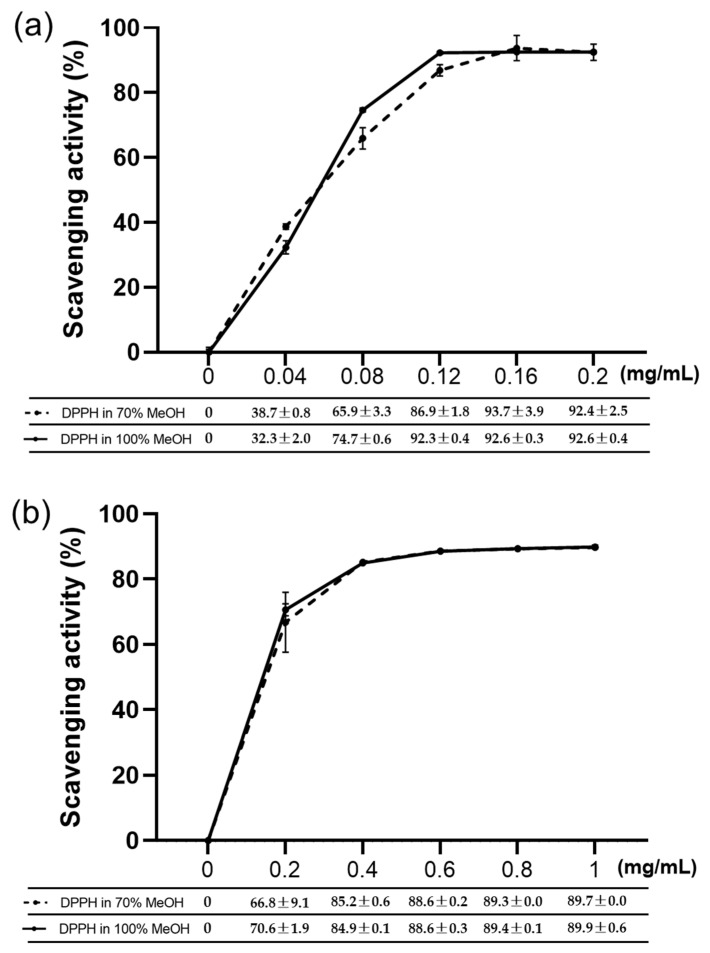
Scavenging activities of standard antioxidants determined by DPPH assays using 70% and 100% aqueous MeOH as the DPPH solvent (mean ± standard deviation, *n* = 3). (**a**) l-ascorbic acid; (**b**) BHT.

**Figure 3 marinedrugs-19-00256-f003:**
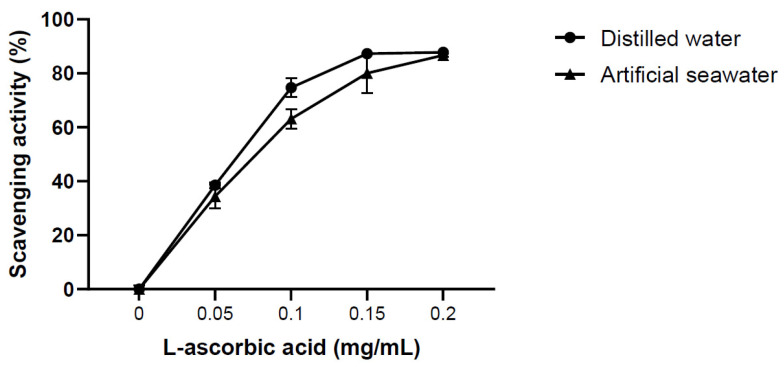
Scavenging activities of l-ascorbic acid solutions in distilled water and artificial seawater (32 psu) (mean ± standard deviation, *n* = 3).

**Figure 4 marinedrugs-19-00256-f004:**
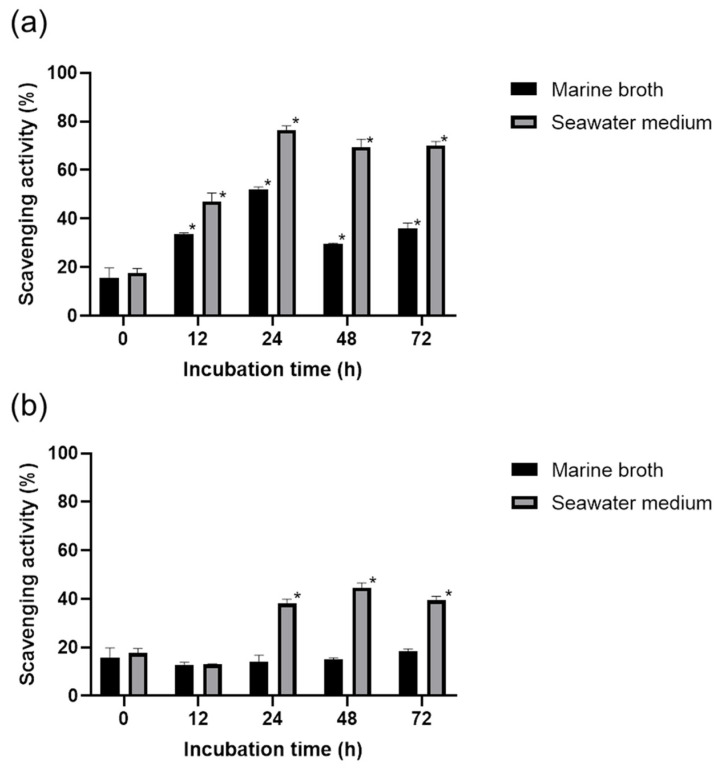
Scavenging activities of the cell-free supernatants of (**a**) *P. rubra* and (**b**) *P. xiamenensis* determined by DPPH assays for two types of media after 12, 24, 48, and 72 h of culturing (mean ± standard deviation, *n* = 3). Asterisks indicate a significant mean difference between the test groups by medium (analysis of variance, *p* < 0.05).

**Figure 5 marinedrugs-19-00256-f005:**
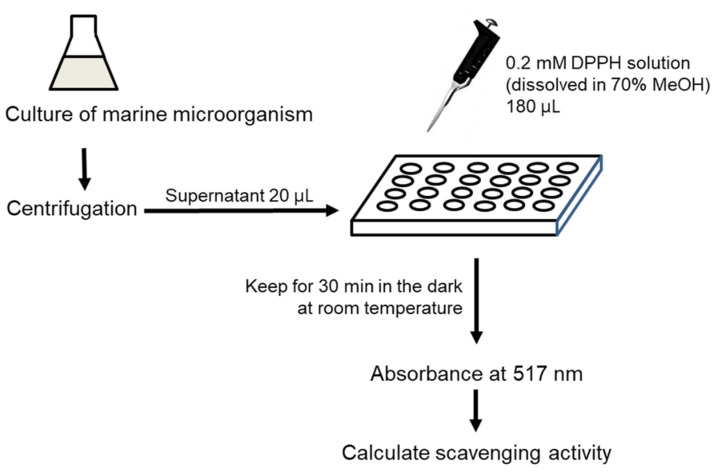
Steps of DPPH assay for cell-free marine microorganism supernatants.

**Table 1 marinedrugs-19-00256-t001:** DPPH assay conditions previously used for microbial antioxidant analysis.

Bacterium	Test Agent	DPPH Solution	References
*Pseudomonas* PF-6	Polysaccharide	0.2 mM in 95% EtOH	[21]
*Edwardsiella tarda*	Polysaccharide	0.008% in 50% MeOH	[8]
*Bifurcaria bifurcata*	Extracts	0.1 mM in MeOH	[22]
*Streptomyces* VITSVK5 spp.	Extracts	1 mM in EtOH	[23]
*Bacillus* sp.	Culture supernatants	0.1 mM in MeOH	[13]
*Enterococcus durans* LAB18s	Culture supernatants	60 μM in MeOH	[11]
*Lactobacillus* sp.	Culture supernatants	0.2 mM ^1^	[9]

^1^ No information on the solvent was provided.

**Table 2 marinedrugs-19-00256-t002:** Effect of DPPH solvent on the aggregation behavior.

	Solvent (%)	MB	Seawater
Methanol	60	–	–
	70	–	–
	80	+	++
	90	++	++
	100	++	++
Ethanol	60	–	–
	70	–	–
	80	++	++
	90	++	++
	100	++	++

“–” no aggregation, “+” weak aggregation, and “++” strong aggregation.

## Data Availability

Data is contained within the article.

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
