# Peer review of "Optimising the DPPH Assay for Cell-Free Marine Microorganism Supernatants"

_marinedrugs, 2021, doi:10.3390/md19050256_

Round 1

Reviewer 1 Report

In Introduction some introductive lines on antioxidant approach and mention related references such as:

Durazzo A. Study Approach of Antioxidant Properties in Foods: Update and Considerations. 02/2017; 6(3):17., DOI:10.3390/foods6030017

The novelty character of paper should be marked.

Major details and a graphical scheme of Methodologies

Generally the results should be better discussed by comparaing them with previous literature data.

Figures 2 and 3 should be better described.

A section Conclusion including limits, advantages, practical applicatins and further indications should be included.

Reviewer 2 Report

Manuscript title: Optimising the DPPH assay for cell-free marine microorganism supernatants

This paper describes the analysis of antioxidant activity using DPPH as the free radical sample to evaluate the antioxidant activity of cell-free marine microorganism supernatants. The results are reliable and validated. Therefore, it provides some useful information to those who are interested in the antioxidant activity analysis by using DPPH. For this reason, I think it is worth to be published on the Journal. However, some changes have to be made before being accepted for publication.

L41 and L42 The author should re-arrange the sentence.

L74 Did the authors apply a commercial equipment for the psu determination? If yes then please mention its make and model number.

L100-L101 Therefore, it is most suited for the screening of microorganism-produced antioxidants.

DPPH is only one of the antioxidant activity analyses for a reference of the antioxidant capability in a specific sample. Therefore, the sentence should be rephrased to a objective view. 

Figure 2.

For clearer presentation, the authors are requested to re-organize the Figure 2 to Table form and give the statistical analysis data.

Round 2

Reviewer 1 Report

Update and proper references on Research approach should be' added in Introduction 

Author Response

Point 1: Update and proper references on Research approach should be' added in Introduction

: We added references and edited the text of our research approach.

: Line 63-71